# Mesopelagic microbial community dynamics in response to increasing oil and Corexit 9500 concentrations

Shahd Aljandal[1], Shawn M. Doyle[1], Gopal Bera[2], Terry L. Wade[1,2], Anthony H. Knap[1,2], Jason B. Sylvan[1]*

1 Department of Oceanography, Texas A&M University, College Station, TX, United States of America,
2 Geochemical and Environmental Research Group, Texas A&M University, College Station, TX, United States of America

* jasonsylvan@tamu.edu

**Data Availability Statement:** Raw data can be accessed at the National Center for Biotechnology Information Sequence Read Archive under Bioproject ID PRJNA715309 and accession

## Abstract

Marine microbial communities play an important role in biodegradation of subsurface plumes of oil that form after oil is accidentally released from a seafloor wellhead. The response of these mesopelagic microbial communities to the application of chemical dispersants following oil spills remains a debated topic. While there is evidence that contrasting results in some previous work may be due to differences in dosage between studies, the impacts of these differences on mesopelagic microbial community composition remains unconstrained. To answer this open question, we exposed a mesopelagic microbial community from the Gulf of Mexico to oil alone, three concentrations of oil dispersed with Corexit 9500, and three concentrations of Corexit 9500 alone over long periods of time. We analyzed changes in hydrocarbon chemistry, cell abundance, and microbial community composition at zero, three and six weeks. The lowest concentration of dispersed oil yielded hydrocarbon concentrations lower than oil alone and microbial community composition more similar to control seawater than any other treatments with oil or dispersant. Higher concentrations of dispersed oil resulted in higher concentrations of microbe-oil microaggregates and similar microbial composition to the oil alone treatment. The genus *Colwellia* was more abundant when exposed to multiple concentrations of dispersed oil, but not when exposed to dispersant alone. Conversely, the most abundant *Marinobacter* amplicon sequence variant (ASV) was not influenced by dispersant when oil was present and showed an inverse relationship to the summed abundance of *Alcanivorax* ASVs. As a whole, the data presented here show that the concentration of oil strongly impacts microbial community response, more so than the presence of dispersant, confirming the importance of the concentrations of both oil and dispersant in considering the design and interpretation of results for oil spill simulation experiments.

## Introduction

On April 20, 2010, the Deepwater Horizon platform exploded resulting in the loss of 11 human lives and the largest oil spill in U.S. history [1]. Approximately, 3.2–4.1 million barrels

numbers SAMN18344485-SAMN18344557. All chemical data are within the paper and its Supporting Information files.

**Funding:** Funding for this work was provided by the American Petroleum Institute, Oil Spill Emergency Preparedness and Response Subcommittee. We thank API's Science and Technology Work Group for their comments on the draft. The decision to incorporate such comments in the manuscript was at the discretion of the authors. The views expressed herein belong solely to the authors and may not represent the views of API or its members. Members of the Technical Advisory Committee (TAC) provided input, review and comment on the approach, assumptions and data inputs as the study progressed.

**Competing interests:** The authors have declared that no competing interests exist.

of crude oil were discharged into the Gulf of Mexico over the next 5 months while efforts were made to seal the leaking wellhead [2,3]. To increase oil remediation rates, reduce the amount of oil reaching the shoreline, and reduce the exposure concentration, chemical dispersants (approximately 1.8 million gallons) were used during the spill [4]. These dispersants facilitate the emulsification and dissolution of large oil drops of oil into smaller droplets to enhance both physical dispersal and microbial biodegradation in seawater with the aim of helping to prevent oil from reaching sensitive coastal environments [5].

In response to an oil spill, indigenous hydrocarbon degrading microorganisms naturally propagate and mineralize oil compounds [6–10]. The character of the microbial response is dependent on the microorganisms present at the location of the oil spill and the composition of the spilled oil. Individual hydrocarbons have different molecular weights and physicochemical properties (e.g., solubility), which influences their fate in the environment. The low molecular weight (LMW) fractions, comprised largely of n-alkanes, are easily separated by evaporation at the surface of the ocean or removed by biodegradation. High molecular weight (HMW) compounds, including iso-alkanes, cycloalkanes, and polycyclic aromatic hydrocarbons (PAHs) are more recalcitrant and are more difficult to remove from the environment [11]. In a hydrocarbon mixture, microbes degrade LMW compounds first, while HMW PAHs are degraded at a slower rate [12–14].

Although chemical dispersants such as Corexit 9500 are designed to enhance the rate of microbial biodegradation, several studies have come to opposing conclusions. Some studies suggest chemical dispersants inhibit hydrocarbon-degrading microbial communities [15–18], some found a neutral effect or no effect at all [19,20], and some found chemical dispersants stimulate hydrocarbon degraders to feed on oil particles, therefore enhancing the biodegradation process [10,21–26]. Dosing methodology is variable in many studies, potentially leading to these inconsistent results. A recent experiment using dispersed oil revealed that biodegradation rate in surface seawater is dependent on oil dosage in an experiment [27]. Those results were an important step forward in our understanding of how to interpret oil spill simulation experiments. If dosage matters in deeper, colder waters, which are more likely to be impacted by a seafloor well blowout such as the DwH, remains unconstrained. Further, it is important to compare in the same experiment the response of oil-only versus dispersed oil and to investigate both the chemical and microbiological response to the different treatments.

Here we sought to determine the impact of exposure of oil or dispersed oil in different concentrations on hydrocarbon chemistry and microbiology in mesopelagic waters. We used three concentrations of dispersed oil spanning three orders of magnitude, 0.2–20% by volume, to bracket expected minimum and maximum environmental concentrations during application. We also tested the microbial response to oil alone and three concentrations of dispersant only. These treatments were used to characterize the chemical signatures of the chemically enhanced water accommodated fraction (CEWAF) and dispersant only mixtures compared to oil only, and to determine their effect on microbial abundance, community composition as assayed via 16S rRNA amplicon analysis, and biodegradation of oil over a six-week period in natural seawater collected from mesopelagic Gulf of Mexico waters.

## Methods

### Sample collection

Offshore seawater was collected aboard the R/V Pelican on 06 August 2018 from a depth of 550 m in the Gulf of Mexico at 27.6924˚N, 93.9380˚W (S1 Fig). Seawater was transferred from the Niskin bottles on the CTD rosette to 20 L Nalgene carboys that were rinsed three times with sample water prior to filling. The carboys were held in the lab of the R/V Pelican at room

temperature and brought back to Texas A&M University on 07 August, where they remained in a walk-in refrigerator at 10 ˚C, the *in-situ* temperature of the water when collected, until experiment initiation on 10 August. No alteration to the water was conducted prior to the experiment.

### Experimental setup

The collected natural seawater was used to setup twenty-four 2L glass bottles. Eight treatments were prepared (S2 Fig): (1) seawater only with no additions (Control) [2], crude oil alone, supplied as a water accommodated fraction (WAF), (3–5) oil dispersed with Corexit 9500, supplied as a chemically enhanced WAF (CEWAF) and prepared in three concentrations: low (~0.2%), medium (~2%), and high (~20%), and (6–8) Corexit 9500 chemical dispersant alone (Corexit), prepared in three concentrations: low (0.0025%), medium (0.025%) and high (0.25%). The concentraiton of Corexit was the same for the low, medium and high treatments of both CEWAF and dispersant alone treatments when accounting for volume of dispersant in CEWAF. These concentrations were selected because the medium concentration yields concentrations similar to what was observed in situ in the Deepwater Horizon deepwater plume [17]. We chose one order of magnitude concetratin higher and lower than this realistic medium concentration to have a high chance of seeing a range of responses by the microbial community. Each treatment was prepared in triplicate bottles for all timepoints and then destructively sampled at the appropriate sampling interval. Preparation of stock WAF and CEWAF was conducted similarly to Kleindienst et al. (2015). Specifically, 1700 mL of filtered and pasteurized seawater was used to mix 300 mL of MC252 oil at 400 rpm for 48 hours to make stock WAF. For the WAF treatment, 1400 mL of the stock WAF was added to 1400 mL of seawater. No settling time was used after mixing for 48 h. Similarly, 1700 mL filtered, and pasteurized seawater was mixed with 300 mL MC252 oil and 30 mL of Corexit 9500A at 400 rpm for 48 hours to make the stock CEWAF. CEWAF was allowed to settle for 1 hour, and subsequently, 3, 30 and 300 mL of the stock CEWAF was added to 397, 370 and 100 mL of filtered and pasteurized seawater followed by addition of 1400 mL of natural seawater collected from GOM for creation of the low, medium, and high concentration CEWAF treatments, respectively. Initial time points (T0) were taken immediately after the treatments were made. The experiments were kept at 10 ˚C in the dark for the duration of the 6-week experiment.

### Chemical analyses

**Hydrocarbons.** For T0 and T3 samples, 1L from WAF and 500mL for CEWAF and Dispersant only were collected for alkane and PAH analysis. For T6, 1L was collected for all three sets of treatments to ensure enough hydrocarbons for detection. Concentrations were normalized to actual amount collected. Samples were preserved with 100 mL dichloromethane (DCM). Prior to analysis, samples were first spiked with aliphatic and aromatic surrogates (d26-$nC_{12}$, d42-$nC_{20}$, d50-$nC_{24}$, and d62-$nC_{30}$ for aliphatic and d8-naphthalene, d10-acenaphthene, d10-phenanthrene, d12-chrysene, and d12-perylene for PAHs). Extraction was done with DCM (total 200 mL) in a separatory funnel. The extracts were reduced to 2 mL in Hexane by evaporating the extract in a water bath at 55˚C. Silica gel columns were used to separate the aliphatic and aromatic fractions: 50 mL of pentane was eluted through the columns to collect the aliphatic fractions, and 50 mL of a 1:1 pentane/DCM mixture was eluted through the columns to collect the aromatic fractions. The collected fractions were then evaporated to final volume of 1 mL in hexane. Finally, GC internal standards (e.g., d54-$nC_{26}$ for aliphatic hydrocarbons and d10-Fluorene and d12-Benzo(a) pyrene for PAHs) were added. Aliphatic hydrocarbons were then analyzed on an Agilent 7890 gas chromatograph with a flame ionization detector

(GC-FID) while PAHs were analyzed on a Hewlett-Packard 6890 gas chromatograph coupled with a Hewlett-Packard 5973 mass selective detector. Further details on temperature program, column used, and quantification methods are described previously [28,29]. Surrogate recoveries for alkane were: D26-C12 (69.8 ± 20.4%), D42-C20 (90.9 ± 21.9%), D50-C24 (95.7 ± 16.2%), D62-C30 (91.6 ± 17.9%). Surrogate recoveries for PAHs analysis were d8-Naphthalene (71.6 ± 17.2%), d10-Acenaphthene (97.3 ± 17%), d10-Phenanthrene (76.1 ± 12.1%), d12-Chrysene (100.4 ± 11%) and d12-Perylene (80.0 ± 10.7%).

## Microbiological Analyses

**Cell and micro-aggregate abundances.**   Subsamples from each bottle were collected at experiment initiation (T0), after three weeks (T3) and after 6 weeks (T6) and preserved for cell counting with 2% formalin (final concentration) and stored at 4 ˚C. Preserved samples were vacuum filtered through black polycarbonate filters (25 mm diameter, 0.2 μm pore-size) and stained with a DAPI mixture containing 80 μL of 50 μg/mL DAPI dye, 280 μL Vectashield mounting medium for fluorescence, and 1540 μL Citiflour AF1 Glycerol/PBS antifade solution. Direct cell counts were performed with an epifluorescence microscope (Zeiss Axio Imager.M2). Individual cells were enumerated at 1000X magnification, microaggregates were enumerated at 400X magnification.

**Molecular biology, DNA sequencing and analysis.**   At time points (T0, T3, and T6), 100–400 mL of each sample were filtered through Supor® 0.2 μm pore-size, 47 mm membrane filters (PALL Corp., Ann Arbor, MI, USA) and then stored at -80˚C until DNA extraction. Total DNA was extracted from experimental filters and a procedural blank filter using FastDNA Spin kits (MP Biomedical, Santa Ana, CA, USA) according to the manufacturer's instructions. The V4 hyper-variable region of the 16S rRNA gene was amplified from the DNA extracts with GoTaq Flexi DNA Polymerase (Promega Corp., Madison, WI, USA) and 515F and 806R barcoded primers containing Illumina MiSeq adapters [30]. Each sample was amplified in 50 μL reactions with the following cycling parameters: 95˚C for 3 minutes, 30 cycles of (95˚C for 45 seconds, 50˚C for 1 minute, 72˚C for 90 seconds), followed by a final elongation step of 72˚C for 10 minutes. PCR products were visualized on a 1.5% agarose gel electrophoresis to assess amplification success. Amplicons were quantified with the QuantiFluor dsDNA System (Promega) and pooled together at equimolar concentration. The final library was then purified using an E.Z.N.A Cycle Pure Kit (Omega Bio-tek, Inc., Norcross, GA, USA). The purified library was sequenced on an Illumina MiSeq platform (v2 chemistry, 2 × 250 bp) at the Georgia Genomics Facility (Athens, GA, USA).

Sequence read curation and processing was carried using the DADA2 package in R [31]. Raw reads were first processed using standard filtering parameters (maxN = 0, truncQ = 2, rm. phix = TRUE, and maxEE = 2). Quality profiles of the forward (R1) and reverse (R2) reads were manually inspected and then reads were truncated to the length after which the distribution of quality scores began to drop: 240bp and 160bp, respectively. Error rates for the filtered and trimmed R1 and R2 reads were calculated using the *learnErrors* function and subsequently used to denoise reads using the DADA2 sample inference algorithm. The denoised R1 and R2 reads, free of substitution and indel errors, were then merged into amplicon sequence variants (ASV) using a global ends-free alignment. Paired reads containing any mismatches in the overlapping region were removed from the dataset. Chimeric ASVs were identified and removed by using the consensus method within the *removeBimeraDenovo* function. As a final curation step, any ASVs of which ≥1% of its reads were from one of the protocol blanks were removed. A consensus taxonomy for each ASV was assigned using the naïve Bayesian classified method of Wang *et al*., 2007 trained on release 128 of the SILVA reference database [32,33]. Downstream ecological analyses of ASVs was performed with a combination of

mothur, phyloseq, and vegan [34–36]. A summary of sequencing statistics is provided in S1 Table.

Raw data can be accessed at the National Center for Biotechnology Information Sequence Read Archive under Bioproject ID PRJNA715309 and accession numbers SAMN18344485-SAMN18344557.

## Results

### Hydrocarbon chemistry

Alkane concentrations were, on average, 1.5X higher in the WAF treatment than the CEWAF Low treatment, but alkane concentrations were ~8-fold and ~105-fold higher in the CEWAF Medium and CEWAF High treatments, respectively, compared to the WAF. $n$C14-$n$C22 alkanes in all oiled treatments (WAF, CEWAF Low, CEWAF Medium, and CEWAF High) decreased at T3 and T6 compared to the previous timepoint (S3 Fig). The percentage decrease was greatest between T0 and T3, but further decreases occurred between T3 and T6.

PAHs with 2 to 6 aromatic rings were found in all oiled treatments at every time point (S4 Fig). Low molecular weight (LMW) PAHs, namely naphthalene and its alkyl homologs, were the most abundant PAHs in all oil containing treatments. This is consistent with their high abundance in Macondo oil [37], where these naphthalenes account for over 50% of the total PAHs. Additionally, 3- and 4- ring compounds were more abundant in CEWAF treatments compared to oil only (WAF) treatment. Naphthalene concentration decreased after 6 weeks in all CEWAF treatments, but the decrease during the first three weeks was much more rapid than during the second three weeks.

$n$-C17/pristane and $n$-C18/phytane ratios are commonly used as chemical indicators of oil weathering by microbial biodegradation. Since pristane and phytane are recalcitrant to biodegradation, comparing them to an easily degraded compound (e.g., $n$-alkanes) that is closest to their carbon numbers ($n$-C17, $n$-C18, respectively) can explain overall microbial degradation of hydrocarbons [38,39]. Across all hydrocarbon-amended treatments, the ratio of $n$-C17/pristane and $n$-C18/phytane decreased from T0 to T6, indicating microbial degradation of $n$-alkanes (Fig 1). In contrast, the unresolved complex mixture (UCM) increased with time in all hydrocarbon-amended treatments. The %UCM was lowest in WAF at T0 and increased with increasing hydrocarbon concentration in the different treatments (WAF < CEWAF Low < CEWAF Medium < CEWAF High). %UCM was highest in CEWAF Medium at T3 but similar across all treatments at T6 (Fig 1).

### Prokaryotic cell and micro-aggregate abundance

At the start of the experiment (T0), total cell abundance ranged between $6.08{\times}10^3$ (CEWAF High) and $4.79{\times}10^4$ (WAF) cells mL$^{-1}$ (Fig 2). After 6 weeks of incubation, cell abundance increased in all treatments expect WAF by one to two orders of magnitude. In the WAF treatment, cell abundance increased after three weeks of incubation (T3) and then declined to initial abundance after 6 weeks. Prokaryotic cell abundance in CEWAF and Corexit treatments increased in direct proportion to the concentration of dispersant used. Cell abundances increased over time proportionately to the concentration of total alkanes in the treatment for WAF and CEWAF treatments (S5 Fig).

Microscopy cell imaging revealed the formation of oil-microbe micro-aggregates in oiled treatments after three weeks (Fig 2). The abundance of these micro-aggregates increased over time in proportion to amount of oil (Fig 2).

 

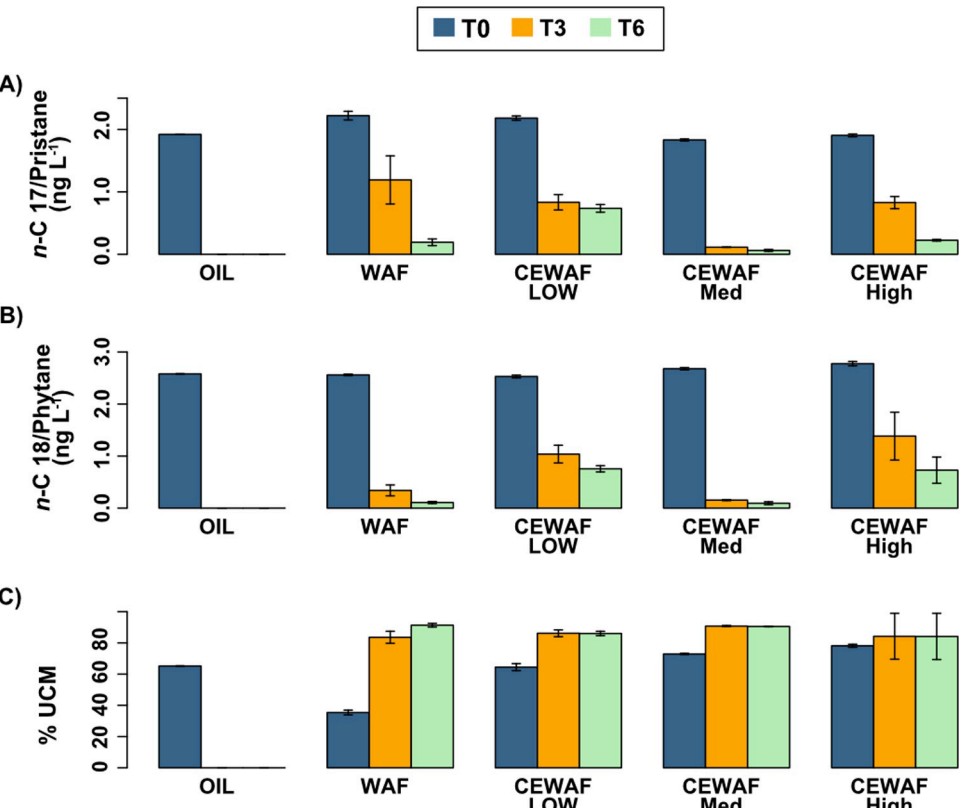

**Fig 1.** Mean ratios of A) *n*-C17 to Pristane, B) *n*-C18 to phytane, and C) %UCM in each treatment for all timepoints during the experiment. Error bars represent standard error +/- mean among replicates (n = 3).

## Microbial community composition and structure

Rarefaction curves for all timepoints after T0 treatments reached saturation, indicating microbial community diversity was exhaustively sampled (S6 Fig). Microbial diversity, as assessed via the inverse Simpson Index (1/D), was statistically the same in WAF and control samples (~2.5± 0.2). In all the Corexit-amended treatments, diversity was higher at T3 and T6 treatments than both the control and WAF treatments (S8 Fig). Within all treatments, diversity was lowest at the end of the experiment, and there was a general trend of decreasing diversity with increasing Corexit concentration.

A principal coordinate analysis (PCoA) of Bray-Curtis dissimilarities was performed to explore how microbial community composition and structure varied between treatments and over time (Figs 3 and S7). At T0 all samples overlapped, confirming the microbial communities within all samples were similar at the start of the experiment. Analysis of similarity (ANO-SIM) was performed to estimate the significance between the microbial communities of each treatment at the initiation of the experiment (T0). The result show no significant difference between the treatments [$R$ = -0.091, $p$ = 0.62].

After three weeks, community composition had diverged into three distinct clusters based on treatment: one with all Corexit only treatments, one with both control and the CEWAF Low treatments, and one with the WAF, CEWAF Medium, and CEWAF High treatments. ANOSIM analysis revealed that there were significant differences in community composition among sampling time points (R = 0.462, S9A Fig) and different treatments (R = 0.225, S9B Fig).

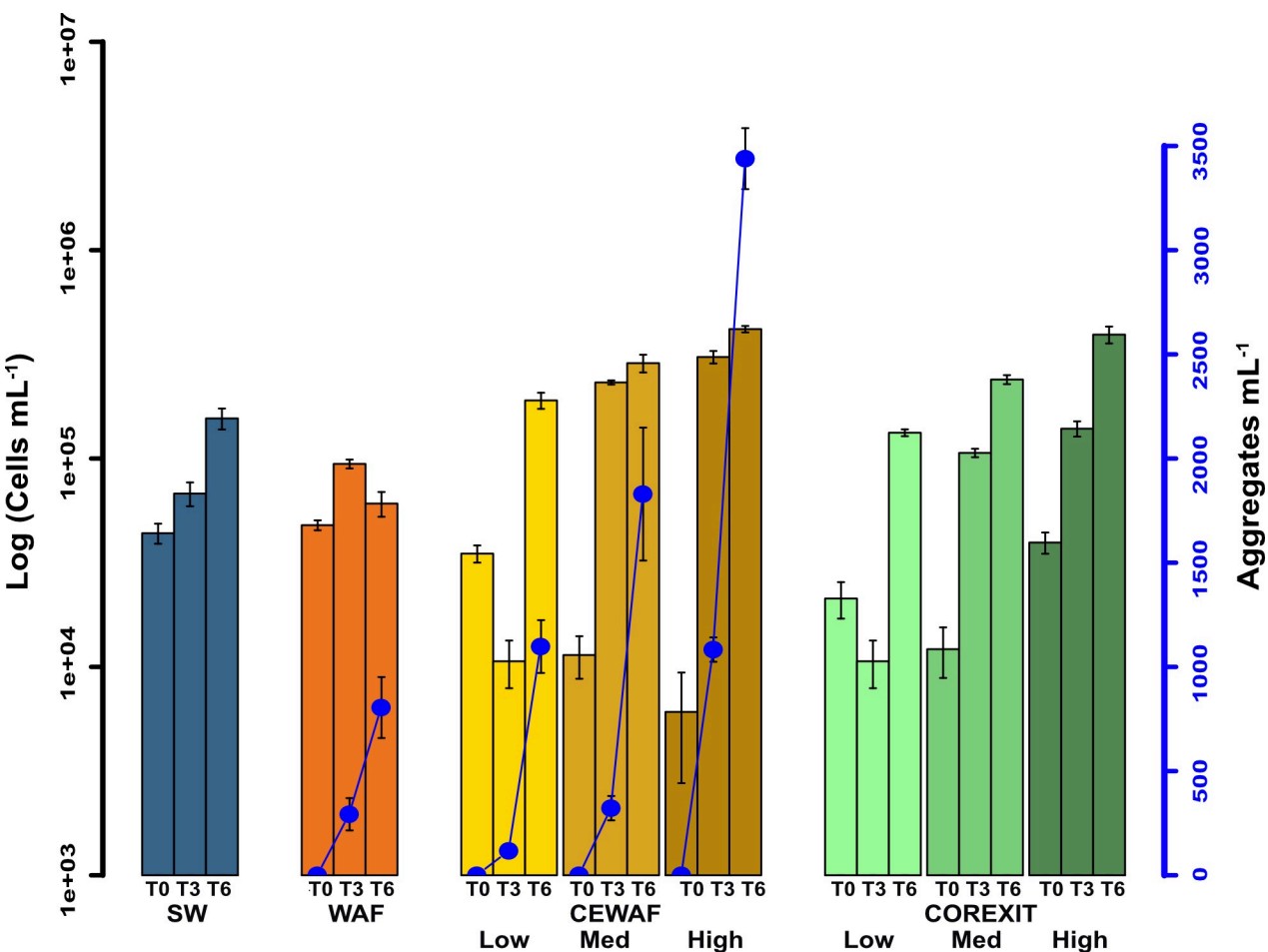

**Fig 2. Prokaryotic cell abundance (bars), and micro-aggregate abundance (dots) observed in each treatment over the 6-week incubation period.**
Error bars represent standard error +/- mean among replicates (n = 3).

The most abundant bacterial lineage in all treatments was *Alteromonadales*, which increased over time in all treatments (Fig 4). The order with the next highest relative abundance was *Oceanospirillales*, which decreased with time in all treatments. The orders *Flavobacteriales* and *Rhodospirillales* increased in relative abundance only in Corexit-amended treatments. The order *Nitrosococcales* was abundant only in T3 and T6 for the CEWAF Low and Control treatments. Finally, *Rhodobacterales* increased in relative abundance in CEWAF Low, CEWAF Medium, and Corexit Low treatments but did not increase in relative abundance in any treatments that were not amended with Corexit. *Nitrosopumilales* appears to dominate the *in-situ* seawater samples.

### Response of putative hydrocarbon-degrading taxa

Analysis of microbial response to the treatments at the ASV level provides more detail to the broad patterns seen at the Order (Fig 4) and genera (S10 Fig) levels. ASV01, classified as *Marinobacter*, was present at the highest relative abundance for all putative hydrocarbon degraders across all timepoints (Fig 5). Its relative abundance increased dramatically after T0 in the WAF, CEWAF Medium, CEWAF High, Corexit Low and Corexit Medium treatments. ASVs 07 and 39 were the next most abundant ASVs classified as *Marinobacter*; ASV07 had highest

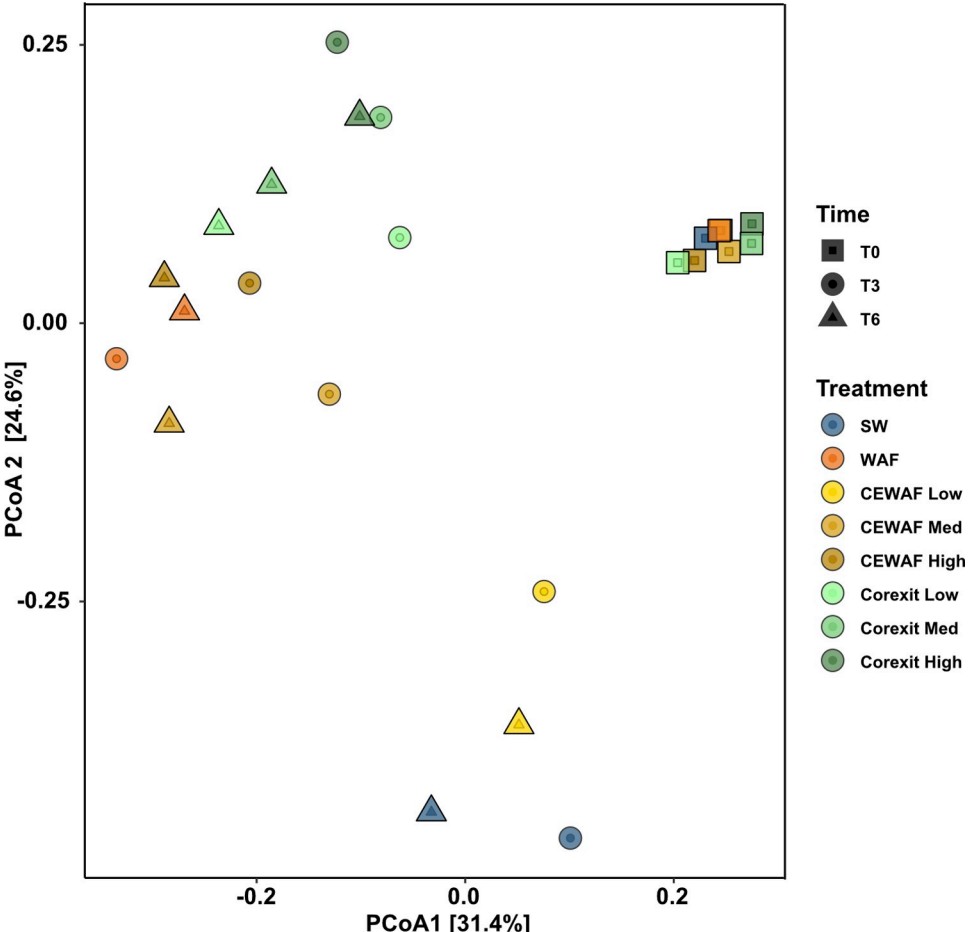

**Fig 3. Principal coordinates analysis (PCoA) of bacterial community composition within each treatment during the 6-week experiment based on weighted-UniFrac distances.** The samples are color-coordinated according to treatment, while the different shapes represent incubation time.

relative abundance in WAF, CEWAF Medium and CEWAF High while ASV39 was most abundant in the Corexit-only treatments, particularly at T0. ASVs 71 and 82, classified as *Cycloclasticus*, a genus known for PAH-degradation, were nearly absent from this experiment except for the WAF treatment at T3 and T6, where ASV71 represented 1.7% of sequences at T3 and ASV82 represented 2.0% of sequences at T6.

ASVs classified as *Colwellia* bloomed only in the CEWAF treatments (Fig 5). At T0, ASVs 14, 25 and 35 each represented <0.05% of the community in any treatment. Combined, these ASVs bloomed to an average of 22%, 11% and 0.15% for CEWAF Low, CEWAF Medium and CEWAF High, respectively, at T3, and 17%, 9% and 0.06% at T6 for the same treatments.

At T0, ASV03, classified as *Alcanivorax*, was the second-most abundant of all putative hydrocarbon degraders (Fig 5). Unlike *Marinobacter*, the relative abundance of *Alcanivorax*, particularly ASV03, significantly decreased across all treatments over time. Its relative abundance was highest when the relative abundance of ASV01 was lowest (S11 Fig). ASV05, classified as *Alcanivorax*, decreased in relative abundance in all treatments except for the WAF treatment, where its relative abundance was consistent at each time point. ASV36, also classified as *Alcanivorax*, was present in all treatments at T0 only except Corexit Medium and Corexit High, where it was nearly absent throughout the experiment.

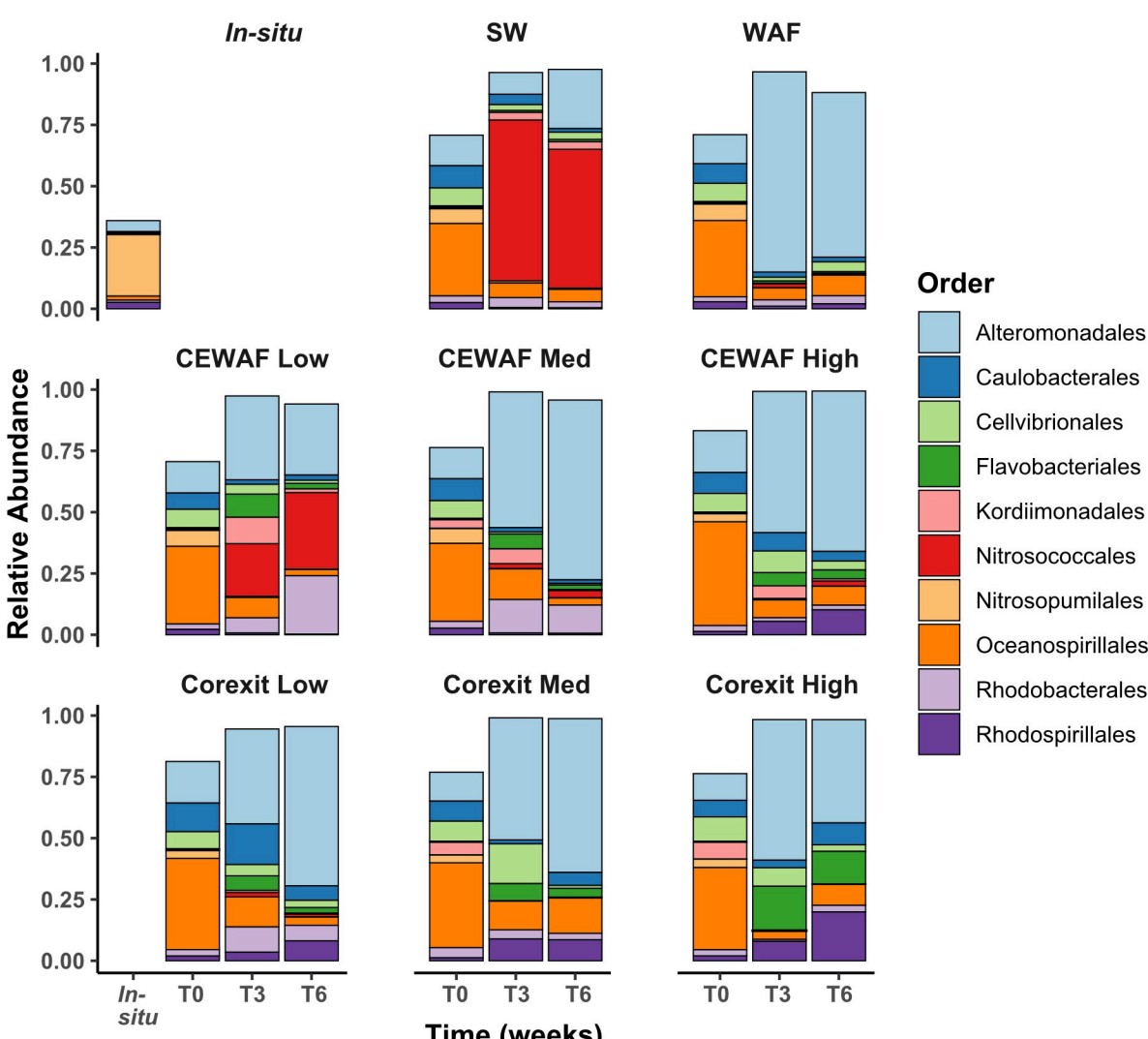

**Fig 4. Relative abundances of top 10 microbial lineages observed *in-situ* seawater samples and each treatment over the 6-week experiment.** Each bar is the average of triplicate treatment.

ASVs classified as *Alteromonas* cannot specifically be associated as putative hydrocarbon degraders without additional genomic data. That said, ASV02 represented 19–50% of the amplicons in the Corexit-amended treatments at T3 and 25–31% at T6 (S12 Fig).

## Discussion

After crude oil is released in the marine environment, chemical dispersants can be applied to mitigate the impact of the spilled oil, including to speed up the rate of oil biodegradation to reduce the toxic influence of hydrocarbons on the marine environment and protect coastal ecosystems. Due to the impact of oil contamination on the environment, many studies have been conducted to understand the degree to which chemical dispersants help improve clean-up responses after such events [17,40,41]. Conflicting results from previous studies have resulted in a debate about the validity of using chemical dispersants to clean up oil spills based on its degree of toxicity. Prior work has shown that different dosing levels can explain some of

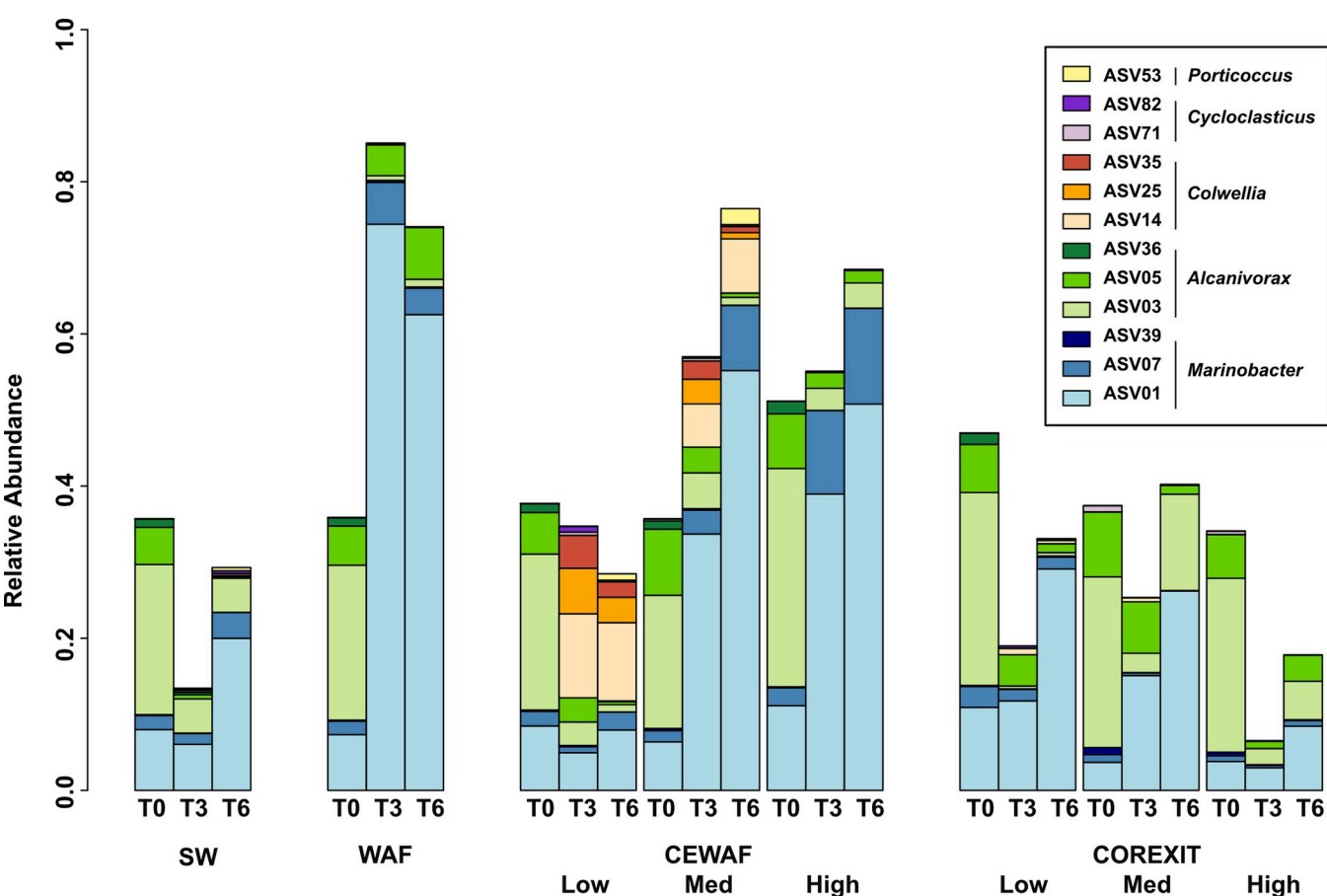

**Fig 5. Relative abundances of selected ASVs belonging to known or putative hydrocarbon degrading genera.** Each bar represents the average of triplicate treatments.

the previous discrepancies for surface seawater [27], where many spills have the greatest impact. However, that work looked only at dispersant amended treatments and focused on oil composition and degradation rate. We here sought to test the hypothesis that dosing level impacts microbial community response in mesopelagic seawater, which is affected in cases of a well blowout such as the Deepwater Horizon and Ixtoc spill, and to look specifically at the microbial community response to different doses of oil with and without dispersant. We found that dosage does impact microbial communities at multiple levels and therefore is important to consider when interpreting experimental results. To our knowledge, this is the first study to explore the dosage effect of chemical dispersants on mesopelagic marine microbial communities.

Alkanes and PAHs decreased overall, which could indicate abiotic or biological degradation. The decreasing ratios of n-C17/pristane and n-C18/phytane over time during this experiment, however, is a clear indication that oil was being degraded by microorganisms in all treatments [28,38,39,42]. Pristane and phytane are used as biomarkers for microbial biodegradation because they are more recalcitrant than n-alkanes with similar number of carbon atoms [42,43]. The most rapid rate and highest amount of biodegradation occurred in the CEWAF Medium treatment (Fig 1). Another indicator of biodegradation, the %UCM, increased over time. Such increases in %UCM are typically observed during biodegradation as lightweight, resolvable compounds are consumed [24].

Microbes vary in their ability to utilize different hydrocarbon compounds found in crude oil. Canonically, simpler components like linear alkanes are degraded first [13,44]. After these compounds are consumed, a succession of different microbes breakdown the more complex HMW components such as aromatics [11,45,46]. Sequencing of the 16S rRNA gene was conducted to examine the effect of different oil with or without different dispersant concentrations on microbial community structure and composition. At T0 all treatments had higher diversity than T3 and T6 (S8 Fig). The observed effect of lower diversity at T3 and T6 is likely related to a combination of the stimulation and blooming of hydrocarbon degrading bacteria in response to hydrocarbon amendment [8] and changes in community composition that can occur in incubation experiments [47]. A decrease in diversity in CEWAF treatments has been detected previously in shorter experiments using surface Gulf of Mexico waters [48,49]. The decrease in diversity in the control treatment indicates that some of the diversity changes were not related to experimental treatments but shifts in hydrocarbon degraders were evident across all treatments, regardless.

In this experiment, microbial communities from different treatments formed distinct clusters in PCoA space (Figs 3 and S7). All treatments at T0 formed a distinct group, which is expected since the same natural microbial community was used as inoculum immediately prior to sampling. The presence of a cluster of samples from control seawater and CEWAF Low treatments at T3 and T6 imply that the amount of oil in CEWAF Low was not enough to drive a strong microbial response. Indeed, the overall oil concentrations in CEWAF Low was lower than in the WAF treatment (S3 and S4 Figs). Both Control and CEWAF Low treatments also experienced an increased relative abundance of *Nitrosococcales* that was not seen in other treatments. Consequently, the CEWAF Low microbial community was more similar to the control than other oil-amended treatments.

In previous studies, the abundance of *Colwellia* was found to peak when simple aromatics increase, indicating that *Colwellia* is capable of aromatic hydrocarbon degradation [50,51]. We observed that *Colwellia* bloomed only in CEWAF Low and, to a lesser extent, in CEWAF Medium (Fig 5). This suggests that the early phase of biodegradation of simple alkanes was completed in these two treatments. It also supports that in CEWAF Low, due to the small amount of bioavailable oil, microbes were able to degrade both simple hydrocarbons and aromatics within 3 weeks. The absence of *Colwellia* enrichment in CEWAF High and lower abundance in CEWAF Medium indicate that more time may be necessary for microbes to finish degrading the alkanes initially present in those treatments where overall oil concentrations were higher.

The amount of dissolved oil in both CEWAF Medium and CEWAF High had similar effect on microbial composition as the WAF treatment, as indicated by their close clustering in the PCoA analysis (Figs 3 and S7). Overall, *Oceanospirillales* and *Alteromonadales* increased in relative abundance in all three of these treatments at T3 and T6, resulting in low Bray-Curtis dissimilarity (0.62 ± 0.2) between them. This could indicate a threshold effect, where a minimal amount of hydrocarbons causes the community to respond, but higher concentrations have less impact. The communities in CEWAF High had elevated relative abundances of putative heterotrophs such as *Flavobacteriales* and *Rhodospirillales*, which were also elevated in the Corexit-only treatments. This indicates that increasing dispersant concentrations eventually favor heterotrophs, who are potentially utilizing carbon in the dispersants and/or byproducts of hydrocarbon remediation.

As expected, microbial communities in the three Corexit-only treatments clustered together, indicating that the presence of dispersant without oil had a similar effect on the initial marine microbial community. Increasing concentrations of dispersant favored *Flavobacteriales* and *Rhodospirillales* at the expense of *Rhodobacterales* and *Caulobacterales*. These Corexit-only responses are similar to those observed in other studies targeting the effect of dispersant only on microbiome composition [17,26].

Our experimental design allowed for analysis of concentration effects of CEWAF and Corexit on individual ASVs. The relative abundance of both ASV01 (*Marinobacter*) and ASV03 (*Alcanivorax*) was high in all treatments, but antagonistic (S11 Fig). Notably, this was less apparent in the Control and Corexit-only treatments, where the outgrowth of ASV01 was less prominent than the oil-amended treatments. It is likely that ASV01 outcompeted ASV03, as indicated by the relative abundance of ASV01 increasing over time at the expense of ASV03. Because ASV01 did not bloom in Control or Corexit-only treatments, it is most likely that ASV01 grew quickly in response to the addition of oil but does not grow as quickly on Corexit alone. *Marinobacter* species are opportunitrophs and will grow quickly in response to new carbon sources [52], whereas *Alcanivorax* ASV03 is likely a slow grower and/or experienced bottle effects. *Marinobacter* ASV07 also appeared unaffected by dispersant concentration in CEWAF but did not have high relative abundance in Corexit-only treatments. This indicates that these important hydrocarbon degraders were likely not impeded by dispersant, although their role in hydrocarbon oxidation specifically cannot be assessed using the methods employed here.

Enrichment of *Colwellia* was notable only in CEWAF treatments, similar to what has been detected in previous work [10,17,41,50,53]. Unlike some previous studies [17,26], we did not detect *Colwellia* in dispersant-only treatments, where we instead found increased relative abundance of *Alteromonas* ASV10, *Flavobacteriales*, *Caulobacterales* and *Rhodospirillales* compared to other treatments. This suggests that the *Colwellia* ASVs detected may have participated in hydrocarbon degradation in our experiments. In support of this, stable isotope probing has shown that *Colwellia* can oxidize ethane and propane, definitively showing that some members of this genus can degrade hydrocarbons. Because Corexit preferentially releases alkanes when mixed with oil, it is possible that *Colwellia* may have biodegraded released alkanes in CEWAF treatments where it was abundant.

In this study, we sought to determine how the dosage of dispersant can impact hydrocarbon chemistry and microbial community response in mesopelagic waters, which are impacted by oil spills released from wellheads at the seafloor. We found that signatures for hydrocarbon biodegradation were related to dosage level, and that the number of oil-microbe microaggregates was proportional to the dosage of dispersant (Fig 2). Importantly, microaggregates were not abundant when dispersant was added without oil, indicating that the increased microaggregate concentrations were related to increased oil concentrations caused by dispersant. Both the presence of dispersant and the dosage level influenced the response of the microbial community. ASVs classified as putative hydrocarbon oxidizing *Marinobacter* were highly abundant in WAF and CEWAF treatments, whereas ASVs of the putative hydrocarbon oxidizer *Colwellia* were abundant only in CEWAF treatments. Some taxa were more abundant in dispersant-only treatments, but it cannot be determined if this was the case due to preference for those treatments or because hydrocarbon oxidizers did not bloom in those treatments, resulting in higher relative abundances for the taxa that were detected. Therefore, the concentration of Corexit matters in determining the rate of biodegradation, and additionally, the concentration determines the length that the experiment should run. Overall, this work confirms that the concentration of dispersant and oil is a critically important consideration when performing oil biodegradation experiments, in general, and highlights the impact of different levels of dispersant or oil-only on specific taxa and how they respond to oil spills over long periods of time.

## Supporting information

**S1 Fig. Sampling site.** Created in R using vector map data from Natural Earth, free vector and raster map data @ naturalearthdata.com. All map data from Natural Earth is in the public

domain and free for use for any personal, educational, or commercial purpose.
(DOCX)

**S2 Fig. Composition of WAF/CEWAF stock and treatments.**
(DOCX)

**S3 Fig. Mean concentrations of n-alkane in oil containing treatments (WAF and all 3 concentrations of CEWAF).**
(DOCX)

**S4 Fig. Mean concentrations of 19 residual PAHs and 22 groups of alkyl-PAHs of oil containing treatments (WAF and all 3 concentrations of CEWAF).** Number of aromatic rings are shown on the upper panel. Note the different y-axis scales for each panel.
(DOCX)

**S5 Fig. The relationship of cell abundance observed within oil-containing treatments (WAF and CEWAF) and the concentration of total alkanes of each treatment.**
(DOCX)

**S6 Fig. Rarefaction curves of each treatment during the 6-week incubation period.**
(DOCX)

**S7 Fig. Principal coordinates analysis (PCoA) of bacterial community composition within each treatment during the 6-week experiment based on Bray-Curtis dissimilarities.** The samples are color-coordinated according to treatment, while the different shapes represent incubation time.
(DOCX)

**S8 Fig. Microbial diversity measured by inverse Simpson Index (1/$D$) for each treatment.**
(DOCX)

**S9 Fig.** Analysis of similarities (ANOSIM) plot showing dissimilarity with R and P values based between and within A) Time Points, B) Treatments. Bold horizontal bar in box indicates median; bottom of box indicates 25th percentile; top of box indicates 75th percentile; whiskers extend to the most extreme data point; width of bar is directly proportional to sample size.
(DOCX)

**S10 Fig. Relative abundances of top 15 microbial genera observed in-situ seawater samples and each treatment over the 6-week experiment.** Each bar is the average of triplicate treatment.
(DOCX)

**S11 Fig. Relative abundance of ASV01 (*Marinobacter*) vs ASV03 (*Alcanivorax*) for all time points and treatments.**
(DOCX)

**S12 Fig. Relative abundances of selected ASVs belonging to *Alteromonas*.** Each bar represents the average of triplicate treatments
(DOCX)

**S1 Table. Number of sequence reads filtered through each step of the DADA2 pipeline.**
(DOCX)

## Acknowledgments

We thank the captain and crew of the R/V *Pelican* for their assistance collecting the seawater sample and getting it back to shore. We thank Kusimica Mitra at Geochemical and Environmental Research Group at Texas A&M University for assistance setting up the experiment and analyzing samples, Brian Buckingham for assistance processing samples during the experiment, and R.J. Wilson for help with experimental design. Yina Liu provided critical review of an early version of this manuscript.

## Author Contributions

**Conceptualization:** Shawn M. Doyle, Gopal Bera, Terry L. Wade, Anthony H. Knap, Jason B. Sylvan.

**Data curation:** Shahd Aljandal.

**Formal analysis:** Shahd Aljandal, Gopal Bera.

**Funding acquisition:** Terry L. Wade, Anthony H. Knap, Jason B. Sylvan.

**Investigation:** Shahd Aljandal, Shawn M. Doyle.

**Methodology:** Shawn M. Doyle, Gopal Bera, Terry L. Wade, Anthony H. Knap, Jason B. Sylvan.

**Project administration:** Terry L. Wade, Anthony H. Knap, Jason B. Sylvan.

**Supervision:** Terry L. Wade, Anthony H. Knap, Jason B. Sylvan.

**Visualization:** Shahd Aljandal.

**Writing – original draft:** Shahd Aljandal, Shawn M. Doyle, Gopal Bera, Jason B. Sylvan.

**Writing – review & editing:** Shahd Aljandal, Shawn M. Doyle, Gopal Bera, Terry L. Wade, Anthony H. Knap, Jason B. Sylvan.

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
