## [Decision Letter · Decision Letter 0]

5 Oct 2021

PONE-D-21-27461Mesopelagic Microbial Community Dynamics in Response to Increasing Oil and Corexit 9500 ConcentrationsPLOS ONE

Dear Dr. Sylvan,

Thank you for submitting your manuscript to PLOS ONE. After careful consideration, we feel that it has merit but needs some revision and clarification. Therefore, we invite you to submit a revised version of the manuscript that addresses the points raised during the review process.

This decision is based on one review, which is both robust and appropriate, and the opinion of this editor. Please address the reviewer's questions and comments prior to acceptance.

We look forward to receiving your revised manuscript.

Kind regards,

Rachel S. Poretsky, PhD

Academic Editor

PLOS ONE

Journal Requirements:

3. We note that Figure S1 in your submission contain map/satellite image which may be copyrighted. All PLOS content is published under the Creative Commons Attribution License (CC BY 4.0), which means that the manuscript, images, and Supporting Information files will be freely available online, and any third party is permitted to access, download, copy, distribute, and use these materials in any way, even commercially, with proper attribution. For these reasons, we cannot publish previously copyrighted maps or satellite images created using proprietary data, such as Google software (Google Maps, Street View, and Earth). For more information, see our copyright guidelines: http://journals.plos.org/plosone/s/licenses-and-copyright.

a) You may seek permission from the original copyright holder of Figure S1 to publish the content specifically under the CC BY 4.0 license.  

Reviewers' comments:

Reviewer's Responses to Questions

**Comments to the Author**

1. Is the manuscript technically sound, and do the data support the conclusions?

Reviewer #1: Yes

2. Has the statistical analysis been performed appropriately and rigorously? 

Reviewer #1: Yes

3. Have the authors made all data underlying the findings in their manuscript fully available?

Reviewer #1: Yes

4. Is the manuscript presented in an intelligible fashion and written in standard English?

Reviewer #1: Yes

5. Review Comments to the Author

Reviewer #1: In the manuscript “ Mesopelagic Microbial Community Dynamics in Response to Increasing Oil andCorexit 9500 Concentrations”, the authors reported the response of natural marine microbial community from the Gulf of Mexico to different concentrations of chemically dispersed crude oil. The authors further analysed the alkane and PAHs concentration dynamics in the different treatments. Although the manuscript is well written and the experiments are sound with good validation, there are two main issues: (1) what are the advantages or innovations of this study compared to previous reported studies apart from the fact that the authors used three different concentrations of dispersants? It would have been more intriguing if multiple dispersants were compared or more advanced ecological analyses were performed to better understand the effects on the microbial populations; and (2) I’m not sure that the results really change much in terms of our view of how oil degradation occurs, and how to make the processes more efficient. There are no surprises that might change the current paradigm.

1. What protocol was used for the preparation of WAF and CEWAF? The mixing speed seems too high (e.g. see Aurand, D. and G. Coelho (Editors). 2005. Cooperative Aquatic Toxicity Testing of Dispersed Oil and the “Chemical Response to Oil Spills: Ecological Effects Research Forum (CROSERF).” Ecosystem Management & Associates, Inc. Lusby, MD. Technical Report 07-03, 105 pages + Appendices).

2. It seems like there was no head space left in the microcosm bottles and therefore, how did the authors ensure that there was adequate oxygenation of the microcosms?

3. How did the authors mix the bottles for the duration of the experiment, and if not, how did they avoid the settling of the crude oil on the water surface?

4. The authors analysed the concentrations of DOSS but no further discussion is provided of why this was done and what are the implications for the microbial community.

In addition, there are a few minor revisions that could be addressed to improve the manuscript.

Line 93: Why was the field water kept at 10C? I assume that this was the temperature of the seawater at the time of collection?

Line 100: What is the rationale for using these concentrations? Please, give more context.

Line 101: Are the low, medium and high concentrations in the Corexit alone treatments the same as in the CEWAF treatments? The supplementary information suggest otherwise but that needs to be explained better.

Line 105: There should be space between “1.4” and “L”

Line 112: Please, specify the duration of the experiment here. You have mentioned it further down but i think this needs to be mentioned in the methods section so that the reader does not end up scrolling up and down to find this information. In addition, three weeks seems a lot of time between sampling points. Previous studies have shown that microbial degradation starts very early on.

Line 115: Can you specify the exact volume of CEWAF and Dispersant only in the treatments?

Line 160: Add space between “5” and “ul”.

Lie 163: Add space between “2.1” and “mm”, “100” and “mm”, and “1.8” and “um”.

Line 190-191: Specify the barcoded primers used for the 16S rRNA amplification.

Line 177: Add space between “25” and “mm”

Line 199: The full stop should be after the citation (33).

Line 223: The “n” in front of C14-C22 should be in Italics. Please, amend the rest of these occurrences in the rest fo the text.

Figure 1. See my previous comment re n-C17, n-C18. Make these consistent throughout the text.

Figure 2: How would you explain the higher number of cells in the SW control at T6 than in the WAF at T6?

Line 268: Delete “oil” in the sentence “…micro-aggregates in oiled treatments oil after three weeks”.

Line 280: It would have been useful to compare the Bray-Curtis PCoA with for example weighted UniFrac as the phylogenetic distance might provide better explanation for the variation between treatments.

Specify in the methods or results how many paired-end sequences were obtained per sample on average and what was the SD between them. What is the number of ASVs after filtration.

Figure 4: It would be more helpful if the figure showed the taxonomic units at family level or show top 15 or 25 genera. Did the authors obtained a taxonomic profile of the in-situ seawater at the time of sampling? This information could be quite useful to better understand how the indigenous community changed when oil/dispersant was added.

6. PLOS authors have the option to publish the peer review history of their article (what does this mean?). If published, this will include your full peer review and any attached files.

Reviewer #1: **Yes: **Christina N. Nikolova, PhD

---

## [Author Response · Author response to Decision Letter 0]

22 Dec 2021

a response to reviewers document was uploaded with the submission, it has detailed responses to the reviewers

---

## [Editor Report · Decision Letter 1]

19 Jan 2022

Mesopelagic Microbial Community Dynamics in Response to Increasing Oil and Corexit 9500 Concentrations

PONE-D-21-27461R1

Dear Dr. Sylvan,

We’re pleased to inform you that your manuscript has been judged scientifically suitable for publication and will be formally accepted for publication once it meets all outstanding technical requirements.

Kind regards,

Rachel S. Poretsky, PhD

Academic Editor

PLOS ONE
---

## [Editor Report · Acceptance letter]

28 Jan 2022

PONE-D-21-27461R1 

Mesopelagic Microbial Community Dynamics in Response to Increasing Oil and Corexit 9500 Concentrations  

Dear Dr. Sylvan:

I'm pleased to inform you that your manuscript has been deemed suitable for publication in PLOS ONE. Congratulations! Your manuscript is now with our production department. 

Kind regards, 

on behalf of

Dr. Rachel S. Poretsky 

Academic Editor

PLOS ONE